Prevalence of text neck posture, smartphone addiction, and its association with neck disorders among university students in the Kingdom of Saudi Arabia during the COVID-19 pandemic

http://orcid.org/0000-0002-6934-5352 Sirajudeen Mohamed Sherif 1 m.sirajudeen@mu.edu.sa
http://orcid.org/0000-0002-4233-9673 Alzhrani Msaad 1
Alanazi Ahmad 1
Alqahtani Mazen 1
http://orcid.org/0000-0003-4737-6833 Waly Mohamed 2
Unnikrishnan Radhakrishnan 1
Muthusamy Hariraja 1
Alrubaia Wafa 1
Alanazi Nidaa 1
Seyam Mohamed K. 1
http://orcid.org/0000-0002-8272-674X Kashoo Faizan 1
Miraj Mohammad 1
Channmgere Govindappa Shashikumar 3
Alghamdi Khalid Ahmed 4
Al-Hussinan Nasser M. 5
1 Department of Physical Therapy and Health Rehabilitation, College of Applied Medical Sciences, Majmaah University , Majmaah , Saudi Arabia
2 Department of Medical Equipment Technology, College of Applied Medical Sciences, Majmaah University , Majmaah , Saudi Arabia
3 Department of Physical Therapy, College of Applied Medical Sciences, University of Hail , Hail , Saudi Arabia
4 Department of Physiotherapy, King Khalid General Hospital , Majmaah , Saudi Arabia
5 Department of Physiotherapy, Hawtah Sudair Hospital , Riyadh , Saudi Arabia
Badicu Georgian
Electronic publication date: 2022 Dec 15
Publication date: 2022
Volume: 10
Electronic Location ID: e14443
Received 2022 Aug 25; Accepted 2022 Nov 1
Copyright: © 2022 Sirajudeen et al.
Copyright year: 2022
Copyright holder: Sirajudeen et al.
License: This is an open access article distributed under the terms of the Creative Commons Attribution License, which permits unrestricted use, distribution, reproduction and adaptation in any medium and for any purpose provided that it is properly attributed. For attribution, the original author(s), title, publication source (PeerJ) and either DOI or URL of the article must be cited.
License URL: https://creativecommons.org/licenses/by/4.0/

Keywords: Smartphone addiction, Neck pain, Text neck, University students, Physical activity, Saudi Arabia

Funding: The deputyship of Research & Innovation, Ministry of Education in Saudi Arabia through the project number (IFP – 2020-24) This research was funded by the deputyship of Research & Innovation, Ministry of Education in Saudi Arabia through the project number (IFP – 2020-24). The funders had no role in study design, data collection and analysis, decision to publish, or preparation of the manuscript.

==============================
The smartphone emerges as an inevitable gadget in modern society and its increased usage results in neck disorders among its users. However, the factors associated with neck disorders among smartphone users are ambiguous and less explored in the literature. The purpose of this research was to determine the prevalence of text neck posture, smartphone addiction/overuse, and its association with neck disorders among university students in the Kingdom of Saudi Arabia during the COVID-19 pandemic. A total of 313 university students who were aged 18 years and older, owned a smartphone, and used it during the preceding 12 months participated in this cross-sectional study. A self-administered questionnaire was used to collect data regarding the prevalence of text neck posture, smartphone addiction/overuse, neck disorders, and the level of physical activity. Binary logistic regression was used to determine the association between the prevalence of neck disorders and text neck posture, smartphone addiction/overuse, and level of physical activity. The 12 months prevalence of neck disorders due to smartphone use among the participants was found to be 46%. The neck disorders were more prevalent among participants who reported text neck posture (P < 0.001) and categorized as smartphone-addicted/overuse (P < 0.001). Measures to promote the awareness of healthy use of smartphones including postural education and to decrease its screen time are warranted to reduce neck disorders.

Introduction

In recent days, there is a steep increase in smartphone use among university students for internet access, social networking, educational purposes, gaming, and other daily life activities (Shah & Sheth, 2018). The smartphone emerges as an inevitable gadget in modern society and its increased usage leads to addiction and other physical problems among users (Porter, 2010; Shaw & Black, 2008; OReilly, 1996). Addiction to the smartphone emerges as a crucial global concern in recent times, especially during the COVID-19 pandemic (Fatima et al., 2021). During the recent COVID-19 lockdown and social isolation, online platforms and web-based tools were used to carry out employment and educational needs. Moreover, people spent a considerable amount of time engaging on social media and networking sites using smartphones which further increased addiction among the users (Caponnetto et al., 2021).

The symptoms of smartphone addiction include a longing for, withdrawal, tolerance, disturbances in daily life, and an inclination towards virtual online community companionship (Kwon et al., 2013b, 2013a). Smartphone addiction and the subsequent overuse were associated with memory and attention problems resulting in a significant reduction in academic performance and health-related quality of life among students (Buctot, Kim & Kim, 2020; Khan, Khalid & Iqbal, 2019; Alkhateeb et al., 2020). Earlier researchers also reported an association between smartphone addiction and musculoskeletal symptoms, eating disorders, and insomnia (Alhazmi et al., 2018; Domoff et al., 2020; Kumar, Chandrasekaran & Brahadeeswari, 2019). The distraction due to smartphone use while driving increases the risk of road traffic accidents and their related consequences (Olsen, Shults & Eaton, 2013). There is a steep rise in the prevalence of smartphone addiction among university students in Saudi Arabia from 19.1% to 60.3% reported in 2016 and 2019 respectively (Alkhateeb et al., 2020; Alsalameh et al., 2019).

Musculoskeletal disorders (MSDs) refer to a wide range of inflammatory and degenerative pathologies involving muscles, tendons, ligaments, joints, nerves, and vascular elements. Pain, aching, burning, stiffness, tingling, and numbness are some common MSD symptoms (Sirajudeen et al., 2018a; Punnett & Wegman, 2004; da Costa & Vieira, 2010). Globally, MSDs emerge as one of the leading causes of disability affecting the activities of daily living, and work capacity resulting in significant social and economic burdens (Leclerc et al., 2014). Early detection and intervention for MSDs are effective in decreasing the disability and the related debilitating consequences (Stover et al., 2007).

Smartphone addiction and overuse are associated with MSDs, especially in the neck and upper limbs. While viewing the smartphone the user flexes the neck to look down at the screen resulting in excessive lordosis in the lower cervical region with a compensatory kyphosis in the upper thoracic region. This faulty posture is referred to as the “forward head”, “turtle neck” or “text neck” posture which due to excessive gravitational moment could abnormally load the articular structures of the cervical spine and neck extensor muscles resulting in inefficiency and fatigue (Derakhshanrad et al., 2021; Damasceno et al., 2018). However, the results of the studies to determine whether inappropriate neck posture leads to neck disorders are ambiguous. Few studies support the hypothesis that faulty neck posture is associated with the occurrence of neck symptoms (Ruivo, Pezarat-Correia & Carita, 2014; Nejati et al., 2014). Whereas few authors did not support the fact that inappropriate posture is a major concern for neck disorders (Grob, Frauenfelder & Mannion, 2007; Kumagai et al., 2014).

The prevalence of neck pain also increased from 39.2% in the year 2016 to 60.8% in 2019 among university students in Saudi Arabia (Alkhateeb et al., 2020; Alsalameh et al., 2019). The earlier research which aimed to study the relationship between smartphone addiction and neck disorders among university students in Saudi Arabia suffers some methodological issues and limitations which are worthwhile mentioning here. The methodological section did not state whether the questions regarding the neck pain inquire whether the symptom was felt or aggravated due to smartphone use. The earlier studies also did not employ a standard case definition of a musculoskeletal disorder encompassing the parameters such as intensity, duration, and frequency of the neck symptoms to discriminate the significant MSDs from the minor ones. Case definitions are a crucial feature of public health surveillance systems (Coggon, 1999). Failure to use case definitions complicates the interpretation of surveillance data and to postulate preventive measures (Hirve et al., 2020; Sirajudeen et al., 2020). Lastly, earlier researchers also fail to include a postural evaluation component.

Addressing the methodological shortcomings and limitations in the earlier research published in the entire Gulf region, the current study aimed to determine the prevalence of neck disorders, text neck posture, and smartphone addiction/overuse among university students in the Kingdom of Saudi Arabia during the COVID-19 pandemic. It is hypothesized that the prevalence of neck disorders among smartphone users may be associated with text neck posture and smartphone addiction/overuse. The findings of this study could trigger an initiative for awareness and strategies to prevent the occurrence of neck pain among smartphone users.

Materials and Methods

Participants

Both male and female students aged between 18–45 years belonging to constituent departments of the college of applied medical sciences, Majmaah University, Kingdom of Saudi Arabia participated in this cross-sectional study. The data collection was performed on the college of applied medical sciences campus between March 2021 and May 2021. The students aged 18 years and above who owned a smartphone and use it in the preceding 12 months were included. The individuals with a history of cervical fractures or surgeries, congenital or acquired musculoskeletal deformities, neurological diseases, or currently pregnant were excluded (Derakhshanrad et al., 2021; Damasceno et al., 2018; Namwongsa et al., 2018).

Ethics statement

The ethical guidelines recommended in the Declaration of the Helsinki (1964) were followed in all the stages of this current research. Majmaah University Research Ethics Committee issued the ethical approval for this study (MUREC-Dec.30/COM-2020/18-2). All the participants signed the written informed consent in English before enrollment in the study. The participant’s privacy and anonymity were protected, and no identifying information was obtained through the study questionnaire. This study did not include any minor participants.

Measurement

The questionnaire used in this study consisted of six sections including socio-demographic, smartphone usage, self-report of text neck posture, standardized Nordic musculoskeletal questionnaire—neck component, smartphone addiction, and physical activity. The sociodemographic section was comprised of items related to age, gender, height, weight, hand dominance, department, and level of education. The body mass index (BMI) was determined by dividing the weight (kilograms) by the height (meter2) of the participants (Louis, 2002). The smartphone usage section consisted of items like duration of smartphones, tablet and laptop computer usage in years, duration of daily smartphone use in general and for a specific purpose like study activities, social media, and playing games, and the postures adopted in holding the smartphones. The data regarding the prevalence of text neck posture, smartphone addiction/overuse, neck disorders, and the level of physical activity were recorded using the self-perception method, smartphone phone addiction scale—short version, standardized Nordic questionnaire-neck component, and international physical activity questionnaire—short form respectively. The authors have permission to use these instruments/tools from the copyright holders.

Self-report of text neck posture

The participants were asked to select the picture that best describes the posture adopted while using a smartphone. Pictures C and D were considered as “Text neck posture” (Fig. 1). Damasceno et al., (2018) reported 91.1% of agreement between test-retest measurements and the k coefficient (Kappa = 0.74, IC 95% 0.54–0.86) indicated as substantial.

Figure 1 Self-perception of neck posture during smartphone use.

Reprinted by permission from Damasceno et al., (2018) Copyright©2018.

Standardized Nordic musculoskeletal questionnaire

The neck component of the Standardized Nordic musculoskeletal questionnaire (SNMQ) was utilized to determine the musculoskeletal disorders in the neck region. SNMQ is a valid and reliable instrument and is widely used in epidemiological studies to screen musculoskeletal symptoms (Kuorinka et al., 1987). The participants recorded the symptoms like pain, numbness, tingling, aching, stiffness, and burning in the neck region which they experienced during or after smartphone use in the preceding 12 months. The participants also reported the intensity/severity, duration, and frequency of the neck symptoms. The musculoskeletal disorder was defined by the experience of the symptoms listed above with moderate pain or more that lasted for a minimum of one-week duration or occurred at least once a month during the preceding 12 months (Bernard et al., 1994). The researchers affiliated with National Institute for Occupational Safety and Health developed and standardized this case definition.

Smartphone addiction scale-short version (SAS-SV)

The SAS-SV developed by Kwon et al. (2013a) was used to determine smartphone addiction/overuse. The SAS-SV comprises 10 self-reported items and is scored on a 6-point Likert scale where “1” represents “strongly disagree” and “6” denotes “strongly agree”. The overall score of SAS-SV ranges from 10 to 60, where the score is directly proportional to the extent of smartphone use in the past year. The psychometric properties of SAS-SV like content and criterion validity and internal consistency (Cronbach’s alpha: 0.91) were found to be adequate. The score of ≥31 and ≥33 denotes smartphone addiction/overuse among males and females respectively (Kwon et al., 2013a). Earlier researchers also employed this cut-off to screen smartphone addiction/overuse among university students (Alsalameh et al., 2019; Baabdullah et al., 2020).

International physical activity questionnaire—short form (IPAQ-SF)

The IPAQ-SF was used to assess the physical activity of the study participants. IPAQ-SQ is a valid and reliable self-report questionnaire consisting of nine items to recall the physical activities performed during the previous 7 days. The data collected using IPAQ-SF are used to determine the participant’s metabolic equivalent task (METs) and categorized as light intensity (less than 3 METs), moderate-intensity (3 to 6 METs), and vigorous-intensity activities (more than 6 METs) (Lee et al., 2011; Craig et al., 2003).

Pretesting

A five-member expert panel consisting of two physical therapists, two orthopedic surgeons, and one public health physician evaluated the comprehensibility of the questionnaire. A sample of 30 university students participated in the pretesting. The members of the expert panel and the participants of the pretesting admit that the questionnaire was clear and easy to understand for university students.

Sample size calculation

The sample size was determined using the Sample Size Calculation for Estimating a Single Proportion method. By considering the prevalence of neck pain among Qassim University students (60%), the required sample size was identified as 277 with 95% confidence and 5% absolute precision (Alsalameh et al., 2019).

Statistical analysis

The data were analyzed using SPSS (version 26.0) for Windows. Descriptive statistics were produced for socio-demographic characteristics, smartphone usage, and prevalence of musculoskeletal disorders of the neck. The prevalence of neck MSDs was determined by dividing the number of participants categorized as neck MSDs based on the case definition by the total number of study participants. The binary logistic regression analysis (Wald Chi-squared test) was used to determine the association between the study variables and the presence/absence of neck disorders among the participants. The statistical significance was set at a 5% of probability level.

Results

A total of 313 students participated in this study. Their socio-demographic characteristics were presented in Table 1. The mean age of the participants was 22.6 years. Most of the participants were female students (54.3%). The mean BMI of the participants was 23.92 Kg/m2. Most of the participants were right-hand dominant (88.2%). More than half of the participants were physical therapy students (53.3%). Most of the participants were bachelor-level students (84.7%). Regarding self-report of physical activity, most of the participants were categorized as light physical activity (45.7%), whereas 39.3% and 15% belonged to moderate and vigorous physical activity categories respectively.

Table 1 Socio-demographic characteristics.

Characteristics	Mean (SD)/Frequency (%)	
Age (Years)	22.6 (±4.08)	
Gender		
Male	143 (45.7%)	
Female	170 (54.3%)	
Height (cm)	164.42 (±10.16)	
Weight (kg)	65.97 (±18.54)	
Body mass index (Kg/m2)	23.92 (±5.13)	
Hand dominance		
Right	276 (88.20%)	
Left	31 (9.91%)	
Both equal (ambidextrous)	6 (1.91%)	
Department		
Physical therapy	167 (53.35%)	
Nursing	51 (16.29%)	
Medical laboratory sciences	29 (9.26%)	
Medical equipment technology	33 (10.54%)	
Medical imaging	33 (10.54%)	
Education level		
Bachelor	265 (84.66%)	
Post graduate/Master	48 (15.34%)	
Physical activity		
Light	143 (45.70%)	
Moderate	123 (39.29%)	
Vigorous	47 (15.01%)	
Note:

SD, Standard deviation.

The study participants reported a mean duration of 9.58 years of smartphone use, 3.97 years of tablet use, and 6.95 years of laptop use. Most of the participants (52.1%) reported using smartphones for 7 h or more daily. One hundred and nineteen participants (38%) used smartphones for less than an hour daily for study purposes. However, 44.7% of the participants spent 4 h or more daily on social media platforms using smartphones. Most of the participants spent less than an hour daily playing games using smartphones. About 55% of the participants reportedly used their right hand to hold their smartphones. Most of the participants (62.6%) reported text neck posture. The prevalence of smartphone overuse/addiction among the participants was 55.3% (Table 2).

Table 2 Smartphone usage of the participants.

Characteristics	Mean (SD)/Frequency (%)	
Smart phone and other gadget usage (years)		
Smart phone	9.58 (±2.66)	
Tablet	3.97 (±3.84)	
Laptop	6.95 (±4.74)	
Daily smartphone use		
About an hour	2 (0.6%)	
1–3 h	22 (7%)	
3–5 h	51 (16.3%)	
5–7 h	75 (24%)	
7 h or more	163 (52.1%)	
Purpose of smart phone use		
Study		
Less than an hour	119 (38%)	
1–2 h	53 (16.9%)	
2–3 h	42 (13.4%)	
3–4 h	38 (12.1%)	
4 h or more	61 (19.5%)	
Social media		
Less than an hour	16 (5.1%)	
1–2 h	36 (11.5%)	
2–3 h	45 (14.4%)	
3–4 h	76 (24.3%)	
4 h or more	140 (44.7%)	
Playing games		
Less than an hour	209 (66.7%)	
1–2 h	36 (11.5%)	
2–3 h	18 (5.8%)	
3–4 h	18 (5.8%)	
4 h or more	32 (10.2%)	
Holding the Smart phone		
Right hand	172 (55%)	
Left hand	19 (6.1%)	
Both hands	118 (37.7%)	
Use of cradle, stand, table or other rest	4 (1.3%)	
Self-report of Text neck posture		
Yes	196 (62.6%)	
No	117 (37.4%)	
Smart phone addiction		
Overuse	173 (55.3%)	
Non-overuse	140 (44.7%)	
Note:

SD, Standard deviation.

The 12 months prevalence of neck disorders due to smartphone use among the participants was found to be 46%. The results of binary logistic regression analysis to determine the association between socio-demographic characteristics and the prevalence of neck disorders were presented in Table 3. None of the sociodemographic parameters was significantly associated with the prevalence of neck disorders. The results of binary logistic regression analysis to determine the association between smartphone usage and the prevalence of neck disorders were presented in Table 4. The neck disorders were more prevalent among participants who reported text neck posture (P < 0.001) and categorized as smartphone-addicted/overuse (P < 0.001).

Table 3 Association between neck disorders and sociodemographic characteristics.

Characteristics	Neck disorders	Significance	Hypothesis test	Odd ratio	
	Yes	No	P value	Wald Chi-Square	df	Unadjusted (95% CI)	Adjusted
(95% CI)	
	Mean (SD)/Frequency (%)	Mean (SD)/Frequency (%)	
Age (Years)	23.03 (±4.48)	22.27 (±3.68)	0.073	3.207	1	1.047
(0.99–1.107)	1.059
(0.995–1.128)	
Gender		
Male	60 (41.9%)	83 (58.1%)	-ref	-ref	-ref	-ref	-ref	
Female	84 (49.4%)	86 (50.6%)	0.075	3.166	1	1.351
(0.863–2.115)	1.883
(0.938–3.780)	
Height (cm)	163.61 (±11.14)	165.12 (±9.21)	0.53	0.395	1	0.985
(0.963–1.008)	0.989
(0.954–1.025)	
Weight (kg)	67.36 (±19.35)	64.79 (±17.79)	0.437	0.604	1	1.008
(0.995–1.020)	1.012
(0.981–1.044)	
Body mass index (Kg/m2)	24.40 (±5.22)	23.51 (±5.03)	0.96	0.003	1	1.035
(0.99–1.081)	1.003
(0.908–1.107)	
Hand dominance		
Right	130 (47.1%)	146 (52.9%)	-ref	-ref	-ref	-ref	-ref	
Left & Both equal (ambidextrous)	14 (37.8%)	23 (62.2%)	0.196	1.675	1	0.684
(0.338–1.384)	0.616
(0.296–1.283)	
Physical activity		
Light	74 (51.7%)	69 (48.3%)	0.369	0.808	1	1.580
(0.81–3.084)	1.372
(0.688–2.734)	
Moderate	51 (41.5%)	72 (58.5%)	0.840	0.041	1	1.044
(0.527–2.069)	0.930
(0.461–1.878)	
Vigorous	19 (40.4%)	28 (59.6%)	-ref	-ref	-ref	-ref	-ref	
Note:

CI–Confidence interval, df–Degree of freedom, SD–Standard deviation.

Table 4 Association between neck disorders and smartphone usage.

Characteristics	Neck disorders	Significance	Hypothesis test	Odd ratio	
	Yes	No	P value	Wald Chi-Square	df	Unadjusted
(95% CI)	Adjusted
(95% CI)	
	Mean (SD)/Frequency (%)	Mean (SD)/Frequency (%)	
Smart phone and other gadget usage (years)		
Smart phone	9.53 (±2.85)	9.62 (2.49±)	0.719	0.130	1	0.988
(0.909–1.074)	0.977
(0.859–1.111)	
Tablet	4.18 (±3.92)	3.79 (±3.78)	0.445	0.584	1	1.026
(0.969–1.088)	1.036
(0.946–1.136)	
Laptop	7.05 (±4.79)	6.86 (±4.71)	0.831	0.045	1	1.009
(0.962–1.057)	0.992
(0.92–1.069)	
Daily smartphone use		
Less than 5 h	26 (34.7%)	49 (65.3%)	-ref	-ref	-ref	-ref		
5 h or more	118 (49.6%)	120 (50.4%)	0.527	0.401	1	1.853
(1.081–3.177)	1.319
(0.56–3.106)	
Purpose of smart phone use		
Study		
Less than 3 h	92 (43%)	122 (57%)	-ref	-ref	-ref	-ref	-ref	
3 h or more	52 (52.5%)	47 (47.5%)	0.073	3.212	1	1.467
(0.909–2.367)	2.004
(0.937–4.286)	
Social media		
Less than 3 h	38 (39.2%)	59 (60.8%)	-ref	-ref	-ref	-ref	-ref	
3 h or more	106 (49.1%)	110 (50.9%)	0.621	0.245	1	1.496
(0.919–2.435)	0.815
(0.362–1.833)	
Playing games		
Less than 3 h	123 (46.8%)	140 (53.2%)	-ref	-ref	-ref	-ref	-ref	
3 h or more	21 (42%)	29 (58%)	0.01*	6.634	1	0.824
(0.447–1.519)	0.324
(0.138–0.764)	
Holding the Smart phone		
Right hand	73 (42.4%)	99 (57.6%)	0.034*	4.054	1	0.714
(0.448–1.137)	0.47
(0.234–0.944)	
Left hand	9 (47.4%)	10 (52.6%)	0.833	0.044	1	0.871
(0.331–2.293)	0.816
(0.122–5.431)	
Both hands/Use of cradle, stand, table or other rest	62 (50.8%)	60 (49.2%)	-ref	-ref	-ref	-ref	-ref	
Self -report of Text neck posture		
Yes	111 (56.7%)	85 (43.3%)	<0.001*	13.385	1	3.324
(2.033–5.436)	3.657
(1.826–7.325)	
No	33 (28.2%)	84 (71.8%)	-ref	-ref	-ref	-ref	-ref	
Smart phone addiction		
Overuse	133 (76.9%)	40 (23.1%)	<0.001*	92.637	1	38.993
(19.173–79.302)	49.553
(22.381–109.710)	
Non-overuse	11 (7.9%)	129 (92.1%)	-ref	-ref	-ref	-ref	-ref	
Notes:

CI–Confidence interval, df–Degree of freedom, SD–Standard deviation.

* Significant (P < 0.05).

Discussion

The results of the current study showed that 46% of the study participants using smartphones experienced neck disorders in the past 12 months. The prevalence rate reported in this study was higher than the earlier study (32.5%) by Namwongsa et al., (2018) among university students in Thailand adopting a similar methodology. The prevalence of neck symptoms among university students using smartphones ranges from 24.2% to 55% (Alsalameh et al., 2019; Al-Hadidi et al., 2019; Chaudary et al., 2019; Almalki et al., 2017; Kim & Kim, 2015). This ambiguity among earlier researchers in reporting the prevalence of neck symptoms may be attributed to methodological differences. Among the earlier studies, only two of them reported 12 months prevalence (Alsalameh et al., 2019; Almalki et al., 2017) and the remaining researchers did not clearly state whether it is 12 months, past week, or point prevalence (Al-Hadidi et al., 2019; Chaudary et al., 2019; Kim & Kim, 2015).

Neck pain was the widely used terminology in the literature (Alsalameh et al., 2019; Al-Hadidi et al., 2019; Almalki et al., 2017; Kim & Kim, 2015). Chaudary et al. (2019) used the terminology “text neck syndrome” and screened the study participants using the neck disability index. Whereas the neck disability index could just provide the disability due to neck pain at that moment and does not have the scope to screen “text neck syndrome” (Al-Hadidi et al., 2019). It is recommended to screen the occurrence of the musculoskeletal disorder based on the parameters like intensity, duration, and frequency of the presenting symptom at the anatomical location to determine the significant cases and exclude the minor ones which is a crucial element in reporting epidemiological studies (Sirajudeen et al., 2018b). Most of the earlier studies among similar populations did not clearly state whether they recorded the details of intensity, duration, and frequency of the presenting symptom (Alsalameh et al., 2019; Chaudary et al., 2019; Kim & Kim, 2015) whereas few studies either collected data regarding the intensity or frequency of the symptoms (Al-Hadidi et al., 2019; Almalki et al., 2017).

The results of the current study showed that 55.3% and 62.6% of the study participants reported smartphone addiction and text neck posture respectively. Both parameters were associated with the prevalence of neck disorders among the participants. The profuse use of smartphones like in cases of smartphone overuse/addiction results in substantial wear and tear in the cervical region of the spinal column, alteration in the cervical curvature, and stability and mobility leading to the occurrence of neck dysfunction (Derakhshanrad et al., 2021). The subjects categorized as smartphone addiction/overuse adopt “flexed neck”, “turtle neck” or “text neck” postures as shown in pictures C and D of Fig. 1. While adopting a healthy ideal neck posture, the weight of the head transmitted via the cervical spine was found to be approximately 10 lbs or 4.5 kgs. However, when the head is sustained in the flexed position like in the case of text neck posture, the weight of the head transmitted through the cervical spine raises exponentially resulting in degenerative changes (Harrison et al., 2002). Moreover, this faulty posture may lead to alterations in the length of the muscles acting in the cervical region resulting in inefficiency due to a compromise in the length-tension relationship (Khayatzadeh et al., 2017; Lin, Wang & Wilkinson, 2022). Hence, it is vital to adopt a healthy posture while using a smartphone. Ergonomic experts recommend keeping the smartphone at eye level to avoid head flexion, using smartphone holders, periodic neck exercises, and finally minimizing the duration of smartphone usage (Derakhshanrad et al., 2021).

The neck symptoms were found to be more prevalent among female participants and those who used smartphones for a longer duration and less common among physically active individuals (Derakhshanrad et al., 2021; Almalki et al., 2017; Kim & Kim, 2015). A similar trend was noted in the results of the present study but the difference in the prevalence rates was not statistically significant. In this study, 49.1% of the female participants reported neck disorders as compared to 41.9% of their male counterparts. The differences in the prevalence rates between the gender may be attributed to underlying gender-related psychological and biological factors. Females exhibit a greater tendency to recognize and report the symptoms compared to males. There are also gender differences regarding musculoskeletal architecture, metabolic functions, and hormonal influences inducing pain-related parameters like perception and threshold (Sirajudeen et al., 2018b; Tittiranonda, Burastero & Rempel, 1999).

The results of this study showed that neck disorders were more prevalent (49.6%) among the participants who reported using smartphones for 5 h or more per day. Kim et al. (2013) reported a decrease in the cervical spine proprioception reflected by the increase in the repositioning error associated with extended duration of smartphone use. Park et al. (2015) reported a significant decrease in the pain threshold of the sternocleidomastoid and upper trapezius muscles in the participants classified as heavy smartphone users (more than 5.4 h of smartphone use per day). Earlier research reported a higher prevalence of neck symptoms among subjects who used laptops and tablets for a longer duration (Jacobs et al., 2011; Thorburn, Pope & Wang, 2021). In the current study, the mean duration of laptop and tablet usage was higher among individuals with neck disorders compared to asymptomatic participants, but this difference is not statistically significant.

In the current study, the prevalence of neck disorders was less common (40.4%) among the participants categorized as vigorous physical activity. The mechanism by which how physical activity was associated with decreased occurrence of musculoskeletal disorders was not explained in the literature. A systematic review by Mansi et al. (2014) reported the effectiveness of physical activity in the prevention of hip fractures and reduction of the neck, shoulder, and lower back pain. Regular physical activity was beneficial in improving bone mineral density and muscle capillary density which could partially support the role of physical activity in reducing the occurrence of musculoskeletal disorders (Warburton, Nicol & Bredin, 2006; Warburton, Gledhill & Quinney, 2001; Mandroukas et al., 1984).

The Kingdom of Saudi Arabia imposed a partial nationwide lockdown, social distancing, and closure of educational institutions as measures to prevent and control the COVID-19 pandemic (Alrashed et al., 2020). Educational institutions utilized online-based learning platforms to teach and assess students during this period (Hosen et al., 2021). Due to social isolation, students tend to use smartphones to virtually connect to the online community through social networking sites and spend considerable time playing phone-based games, browsing internet sites, watching social media, and so on. The increased time spent on the smartphone has negative consequences for users being addicted to the smartphone (Ratan et al., 2021). Smartphone addiction/overuse was associated with a decline in academic performance, musculoskeletal pain, poor sleep, stress, anxiety, and negative emotions among university students (Alotaibi et al., 2022). A study by Hosen et al. (2021) reported an alarming level of 86.9% of smartphone addiction among Bangladesh students during this COVID-19 pandemic. The prevalence of smartphone overuse among the participants of the current study (55.3%) is slightly lesser compared to the rates reported among university students in the Jeddah (63%) and Makkah region (67%) in Saudi Arabia reported during the COVID-19 pandemic (Alotaibi et al., 2022; Alsiwed et al., 2021). Taking into consideration the rate of smartphone addiction among university students in Saudi Arabia during the year 2019 (60.3%) which was just before the start of the COVID-19 pandemic and rates reported during the COVID-19 pandemic, it could be inferred that the prevalence of smartphone addiction among university students in Saudi Arabia did not increase during the COVID-19 pandemic period (Alsalameh et al., 2019).

To the best of our knowledge, this is the first study in the entire Gulf region to determine the prevalence of neck disorders by employing a case definition and administering a postural evaluation component. The findings of the current study supported the biomechanical hypothesis that inappropriate neck posture (text neck) may be a cause for the increased prevalence of neck disorders in this population. Moreover, neck disorders were more prevalent among students categorized as smartphone addiction/overuse. Failure to manage one’s leisure time is a risk factor for smartphone addiction (Gezgin, Mihci & Gedik, 2021). In a recent systematic review, Liu (2021) reported that active engagement in physical exercise, sports, and social activities during leisure time are shown to be effective in the management of smartphone addiction/overuse among students.

Limitations

The cross-sectional nature of the study design exercised in the current research could not establish a causal relationship between the associated variables and neck disorders. The data obtained from the participants were self-reported and inherited the risk of recall bias. The convenience sampling technique employed in this study limits us from generalizing the findings to the entire of Saudi Arabia.

Conclusions

The current study’s findings showed that nearly half of the participants experienced neck disorders. Smartphone addiction and text neck posture were significantly associated with the occurrence of neck disorders. Measures to promote the awareness of healthy use of smartphones including postural education are recommended. Active engagement in physical exercise, sports, and social activities during leisure time would be beneficial in decreasing smartphone/overuse and its consequences on neck symptoms among university students. Future studies addressing the effectiveness of measures to promote neck postures and reduce smartphone addiction/overuse and its subsequent alleviation in neck symptoms are warranted.

Supplemental Information

Supplemental Information 1 Data set.

Click here for additional data file.

Supplemental Information 2 Raw data codebook.

Click here for additional data file.

Additional Information and Declarations

Competing Interests

Author Contributions

Human Ethics

Data Availability

Faizan Kashoo is an Academic Editor for PeerJ.

Mohamed Sherif Sirajudeen conceived and designed the experiments, performed the experiments, analyzed the data, prepared figures and/or tables, authored or reviewed drafts of the article, and approved the final draft.

Msaad Alzhrani conceived and designed the experiments, authored or reviewed drafts of the article, and approved the final draft.

Ahmad Alanazi conceived and designed the experiments, authored or reviewed drafts of the article, and approved the final draft.

Mazen Alqahtani conceived and designed the experiments, authored or reviewed drafts of the article, and approved the final draft.

Mohamed Waly conceived and designed the experiments, performed the experiments, analyzed the data, prepared figures and/or tables, and approved the final draft.

Radhakrishnan Unnikrishnan performed the experiments, prepared figures and/or tables, and approved the final draft.

Hariraja Muthusamy performed the experiments, prepared figures and/or tables, and approved the final draft.

Wafa Alrubaia performed the experiments, prepared figures and/or tables, and approved the final draft.

Nidaa Alanazi performed the experiments, prepared figures and/or tables, and approved the final draft.

Mohamed K. Seyam conceived and designed the experiments, authored or reviewed drafts of the article, and approved the final draft.

Faizan Kashoo performed the experiments, prepared figures and/or tables, and approved the final draft.

Mohammad Miraj conceived and designed the experiments, authored or reviewed drafts of the article, and approved the final draft.

Shashikumar Channmgere Govindappa performed the experiments, authored or reviewed drafts of the article, and approved the final draft.

Khalid Ahmed Alghamdi performed the experiments, authored or reviewed drafts of the article, and approved the final draft.

Nasser M. Al-Hussinan performed the experiments, authored or reviewed drafts of the article, and approved the final draft.

The following information was supplied relating to ethical approvals (i.e., approving body and any reference numbers):

The Majmaah University Research Ethics Committee approved the study (MUREC-Dec.30/COM-2020/18-2).

The following information was supplied regarding data availability:

The raw measurements are available in the Supplemental Files.

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
