# Peer review of "Prevalence of text neck posture, smartphone addiction, and its association with neck disorders among university students in the Kingdom of Saudi Arabia during the COVID-19 pandemic"

_PeerJ, doi:10.7717/peerj.14443_

## Round 0.1 · original submission · Major Revisions

Thank you for submitting the manuscript to PeerJ. It has been reviewed by experts in the field and we request that you make major revisions before it is processed further.

We look forward to hearing from you soon.

Best wishes,

Badicu Georgian, Ph.D

·

Basic reporting

Authors reported on a very interesting study on the prevalence of text neck posture, smartphone addiction/overuse, and its association with neck disorders among university students in the Kingdom of Saudi Arabia during the Covid-19 pandemic. The area of the research is interesting; however, it needs a few amendments. Overall, the paper is well-written and well-organised.

Abstract
• Authors should consider adding more info (i.e., the background).
• The info should be better presented as to raise the interest of the readers.
• There are no keywords.

Introduction
• The information given by the authors is not sufficient to create the background for this important matter to health. Can the authors maybe provide some more background on the importance of discovering these effects earlier in life? A paragraph will be fine.
• The rationale of the study should be mentioned more clearly in this section. At least the reader should be informed on the novelty of this study.
• The article innovation should be presented in the Introduction. Describe what the research gap of the paper is and what is new. Please describe the links between the research gap and the goal of the article.
• The hypothesis/research question is not defined.

Experimental design

The experimental design meets the scope of the journal and is clearly explained.
Methods are described detailed enough.

Validity of the findings

Most of the results are quite interesting and are well discussed.
In the Discussion it would be better to have seen more use of terms like 'originality' and 'significance'. Identify what is new in this study that may benefit readers or how it may advance existing knowledge or create new knowledge on this subject. There should be a clear conclusion on why the research findings are significant.

The quality of the images is good enough, but I don’t know if the reviewing version has lower resolution than the final version. If not, images should have better resolution in its final size.

It seems that the English is clear, but research articles usually do not use the word "we/our" and regularly use passive verbs.

·

Basic reporting

An excellent research. I congratulate the authors for presenting a study that offers a solution to a problem that will appeal to a wide range of readers. I think that the large number of participants also increases the quality of the research. In the discussion part of the research, I recommend mentioning the limitations and the practical applications presented.

Experimental design

Perfect

Validity of the findings

Good

·

Basic reporting

Interesting actual problems of using the electronic device for communications, social life and for studding.

Experimental design

It is correct cross-sectional study

Validity of the findings

One hundred and nineteen participants - better 119 participants
graphic of polygon of frequency is more effective for Age, Body mas, Body high, BMI index
Also, for that, continuous variables, for comparing recommend Student t-test, As you write
"The binary logistic regression analysis (Wald Chi-squared test) was used to determine the association between the study variables and the presence/ absence of neck disorders among the participants" for comparing categories we can use Chi-squared test.
Graphics for presenting difference of results can be bar chat. It is more efficient..
If we dont have any student on program Public Health 0%, we can put out that categories.
In Conclusion we must present the mean results with some figures, like in abstract.

Reviewer 4 ·

Basic reporting

No comment.

Experimental design

Adequate.

Validity of the findings

The only concern I have regarding the validity of the findings is that there is no accounting for 'non-smartphone' i.e. tablet and laptop usage. I would imagine students would work mostly on a laptop/tablet to perform reading/homework vs. their cell phone. It is unclear from Table 3 whether this includes laptop use in the answers. Please specify and add discussion to this difference of computer use; if laptop/tablet usage data was not ascertained, then again it should be specifically specified.

Additional comments

Introduction:
Please define 'text neck posture' in the introduction

Lines 29-33 - This sentence is awkward, please change.

Please correct minor edits:
Line 28: Change 'own' to "...owned..."

Line 85: Change 'own' to "...owned..."

Line 96: Change 'such' to "...including..."

Line 104: Change 'patterns adapted' to "...postures adopted..."

Line 111: Change 'adapted' to "...adopted..."

Line 120: Change 'after the smartphone' to "...after smartphone..."

Line 130: Change 'represent' to "...represents..."

Line 130: Please add reference regarding Kwon et al.

Line 221: Please add reference for flexed head position causing degenerative changes
(e.g. Harrison DE, Jones EW, Janik TJ, Harrison DD. Evaluation of axial and flexural stresses in the vertebral body cortex and trabecular bone in lordosis and two sagittal cervical translation configurations with an elliptical shell model. J Manipulative Physiol Ther. 2002 Jul-Aug;25(6):391-401. doi: 10.1067/mmt.2002.126128. PMID: 12183697.)
Line 223: Please add reference for faulty neck posture causing inefficiency due to compromise in the length-tension relationship of the muscles
(e.g. Khayatzadeh S, Kalmanson OA, Schuit D, Havey RM, Voronov LI, Ghanayem AJ, Patwardhan AG. Cervical Spine Muscle-Tendon Unit Length Differences Between Neutral and Forward Head Postures: Biomechanical Study Using Human Cadaveric Specimens. Phys Ther. 2017 Jul 1;97(7):756-766. doi: 10.1093/ptj/pzx040. PMID: 28444241;
Lin G, Wang W, Wilkinson T. Changes in deep neck muscle length from the neutral to forward head posture. A cadaveric study using Thiel cadavers. Clin Anat. 2022 Apr;35(3):332-339. doi: 10.1002/ca.23834. Epub 2022 Jan 25. PMID: 35038194; PMCID: PMC9304288.)

---

## Round 0.2 · accepted · Accept

The paper has been Accepted for publication. Congratulations!

·

Basic reporting

Thank you for providing this comprehensive work.
The authors have presented an improved version of the manuscript.

The introduction provides a proper background of the topic. The sections and the title have been improved. Relevant results are well-organised to follow the hypothesis.
The quality of the images is good enough.
It seems that the English is technically correct.

Experimental design

The experimental design meets the scope of the journal, and it is relevant to the community.
Methods are described detailed enough.

Validity of the findings

The results and the conclusions are quite interesting and well-discussed. All data are provided.

The authors have adequately addressed all my comments. I have no further suggestions.